# Performance and Cost-Efficiency of Single Hormonal Treatment Protocols in Tropical Anestrous Dairy Cows

**DOI:** 10.3390/ani14111564

**Published:** 2024-05-25

**Authors:** Thitiwich Changtes, Javier Sanchez, Pipat Arunvipas, Thitiwan Patanasatienkul, Passawat Thammahakin, Jiranij Jareonsawat, David Hall, Luke Heider, Theera Rukkwamsuk

**Affiliations:** 1Department of Large Animal and Wildlife Clinical Science, Faculty of Veterinary Medicine, Nakhon Pathom 73140, Thailand; changtes.th@gmail.com (T.C.); fvetpia@ku.ac.th (P.A.); passawat.th@vetmed.hokudai.ac.jp (P.T.); 2Department of Health Management, Atlantic Veterinary College, University of Prince Edward Island, Charlottetown, PE C1A 4P3, Canada; jsanchez@upei.ca (J.S.); thitiwan.patanasatienkul@gmail.com (T.P.); lcheider@upei.ca (L.H.); 3Kasetsart University Veterinary Teaching Hospital, Nong Pho, Ratchaburi 70120, Thailand; softonion@gmail.com; 4Faculty of Veterinary Medicine, University of Calgary, Calgary, AB T2N 4N1, Canada; dchall@ucalgary.ca

**Keywords:** dairy cow, anestrous, cost-effective, hormone treatment, reproductive performance

## Abstract

**Simple Summary:**

Simple Summary: Hormonal treatment seems less effective than expected, and it is not a cost-efficient option for solving an anestrous problem in dairy cows under field conditions in tropical countries. To assess whether this evidence is correct, this retrospective study retrieved the clinical records of anestrous cows that received hormonal treatment before artificial insemination (AI) from the Veterinary Teaching Hospital. The results showed that pregnancy success was influenced by hormone treatment protocol, body condition score (BCS) upon hormone treatment, housing conditions, and season. The fixed-time AI (TAI) is the effective method for treating both cyclic and noncyclic anestrous dairy cows due to the high pregnancy risk (PR). However, the cost-efficiency analysis showed that the hormone treatment protocols with estrus detection before AI (EAI) were the most cost-efficient option for cyclic cows. In contrast, the TAI protocols were only appropriate for noncyclic cows. This study emphasized that hormonal treatment is still an effective technique for improving reproductive efficiency in anestrous cattle. However, farmers and veterinarians in tropical countries should be aware of the animal’s health, environmental conditions, farm and management conditions, and the cost-efficiency of treatment protocols before making the treatment decision.

**Abstract:**

This retrospective study aimed to evaluate the performance of hormone treatment protocols, determine the factors associated with pregnancy success after hormone treatment, and compare the cost-efficiencies of two types of hormone treatment among cyclic and noncyclic anestrous dairy cows. The clinical records of 279 anestrous cows that received hormone treatment for artificial insemination (AI) from 64 herds in the western region of Thailand were obtained from Kasetsart University Veterinary Teaching Hospital from January to August 2017. The performance of the hormone treatment protocols, fixed-time AI (TAI) and estrus detection before AI (EAI), showed that the pregnancy risk for the TAI protocol was higher than that for the EAI protocol, but pregnancy per AI did not differ significantly between the two protocols in cyclic and noncyclic cows. Multivariate logistic regression analysis showed that cows receiving the TAI protocol were more likely to be pregnant compared to those treated with the EAI protocol. Cows with a 3.00 body condition score (BCS) < 3.75 after treatment and loose-housed cows were more likely to become pregnant. Treatment during winter showed higher pregnancy success than that in the summer and rainy seasons. The cost-efficiency analysis showed that the TAI protocol was the most cost-efficient option for noncyclic cows, whereas the EAI protocol was the most cost-efficient option for cyclic cows.

## 1. Introduction

Anestrous is one of the most common reproductive problems in crossbred dairy cows in tropical countries. Prevalence estimates range from 20 to 50% [1,2] and result in subsequent financial losses [3]. Anestrous is a multifactorial problem related to negative energy balance (NEB) during the transition period, heat stress, reproductive pathology, and low heat detection efficiency [4]. Farm management interventions such as dietary management after calving [5], reducing heat stress [6], and improving estrus detection practices [7] are commonly used to mitigate these factors.

The bovine estrous cycle is regulated by endogenous hormones produced by the hypothalamic–pituitary–gonadal axis. Exogenous reproductive hormones or their analogs (e.g., GnRH, PGF_2α_, eCG, and progesterone) are commonly used to induce cyclicity in anestrous cows and help reproductive management in dairy herds. Various programs that use these hormones have been developed and can be divided into (1) estrus detection-based programs before artificial insemination (EAI), (2) fixed-time AI (TAI), and (3) combinations of EAI and TAI. The primary purposes of these hormone interventions among anestrous dairy cows are to increase the probabilities of estrous detection and insemination and to increase the probability of pregnancy within a desired period [8]. The probability of pregnancy per cow eligible to be inseminated within the desired after-treatment period is called pregnancy risk [9]. The terms ‘pregnancy risk’ and ‘pregnancy rate’ were discussed in terms of appropriate terminology because they were not evaluated and expressed per unit of time [9]. Therefore, the pregnancy rate could refer to pregnancy risk in some previous studies [5,10,11]. Using hormone treatment can increase pregnancy risk among anestrous dairy cows [5,10]. Moreover, the TAI protocols allow all animals to be artificially inseminated without reliance on heat detection and may be helpful in situations where heat detection efficiency is lacking or when estrous behavior is decreased [5].

Pregnancy risk, after hormone treatment, has been shown to vary between different protocols [11] and is also affected by several animal-related factors, such as the presence or absence of a corpus luteum (CL) at the initial time of hormone implementation, body condition score (BCS), parity, and days in milk (DIM) [12]. Environmental factors such as heat stress influence pregnancy risk [13]; pregnancy risk during the hot and humid season is typically lower than during the cool season [14,15]. Farm management factors such as estrus detection efficiency and proper AI practices also influence pregnancy risk [16,17].

Estrus induction and TAI have been shown to minimize estrus detection and insemination problems on dairy farms [8]. Some farmers question the value of hormone treatment due to its higher cost than estrus detection followed by AI alone. The use of reproductive hormones, synchronization programs, and TAI in anestrous bovine species (cows and buffaloes) in tropical countries has been studied and showed desired results in improving reproductive performance under tropical conditions [11,18,19,20,21,22]; however, information on the cost-efficiency of different hormonal treatment protocols is still limited.

Although several studies have been published evaluating the effect of hormonal treatments on anestrous bovines in recent years, most studies are experimental studies comparing different protocols under the selected conditions or cohort studies on commercial farms in temperate regions. These may not reflect the real-world situation in tropical countries when veterinarians apply treatment protocols on small-holder and semi-commercial farms where the performance of hormone treatment protocols might be affected by the factors mentioned above and veterinarian and farmer decisions for treatment.

Moreover, climate change is more likely to negatively affect the productivity, reproductive performance, and health of livestock in tropical countries and is becoming even more extreme than anyone predicted over the past 10 years. High environmental temperature has unfavorable effects on reproductive function among dairy cattle [23]. Effective cooling systems, housing management, hormone treatment, and other reproductive management technologies have been applied to diminish climatic impacts on reproductive performance. However, anecdotal evidence showed that hormonal therapy appeared to be less effective than expected under field conditions based on discussions with local Thai dairy veterinarians. It is documented that intensive hormonal treatment could improve the reproductive performance of dairy cows and could provide more profits to farmers than less intensive treatment [24]. The high cost of treatment and unpredicted results are important barriers that limit the adoption of hormonal treatment practices for solving the anestrus problem in dairy cows, especially among smallholder dairy producers.

The present retrospective study is the first epidemiological study on hormone treatment protocols in anestrous dairy cattle in Thailand using clinical records from a veterinary teaching hospital. This study aimed to help farmers and clinicians in tropical countries make the most effective decision about hormone treatment in field conditions. The objectives of this study were as follows: (1) to evaluate the performance of hormone treatment protocols on pregnancy risk; (2) to determine factors associated with pregnancy success after hormone treatment; and (3) to assess the cost-efficiency of hormone treatment protocols among tropical anestrous dairy cattle in Thailand.

## 2. Materials and Methods

The study is a retrospective cohort study carried out by the Herd Health and Production Service (HHPS) Unit from the Large Animal Clinic of the Kasetsart University Veterinary Teaching Hospital, Nong Pho. This study was part of a project that aimed to improve cattle reproductive efficiency for small farmers in three provinces of the central-western region of Thailand (that is, Ratchaburi, Kanchanaburi, and Nakhon Pathom). Clinical records of anestrous dairy cows that received hormone treatments were obtained from the HHPS database from January to August 2017.

### 2.1. Dairy Farms and Reproductive Management

Ratchaburi, Kanchanaburi, and Nakhon Pathom are in the tropical climate zone between 13°50′ and 14°10′ N and between 99°50′ and 100°00′ W. The average temperature ranges from 21 °C to 36 °C, and the average relative humidity ranges from 69% to 79% [25]. It is a typical agricultural area containing 13% of the total population of dairy cows in Thailand [26].

Thai crossbred Holstein dairy cattle, commonly implying roughly 87.5% Holstein–Friesian and 12.5% indigenous Thai crossbred dairy cattle [27,28], is the main population among participating dairy farms. The size of the herd was relatively small, ranging from 5 (small-holder herds) to 90 lactating cows (semi-commercial herds). The voluntary waiting period was generally set at 50–60 days. AI was the primary breeding method used. Estrus detection was performed on the basis of visual observation at least twice a day. The AI was usually carried out within 12–16 h after standing heat was observed. Regarding reproductive management practices, the farms had monthly reproductive health checks performed by the Large Animal Unit, including diagnosis and treatment of reproductive disorders, reproductive cycle status and estrus monitoring, and pregnancy diagnosis. The pregnancy diagnosis was made by ultrasound or rectal palpation approximately 35 days after AI.

### 2.2. Data Management

Anestrous cases were defined as the absence of visible estrus detected by farmers for ≥30 days before the day of hormone treatment and diagnosed by clinicians using either ultrasound or rectal palpation. Furthermore, the anestrous cases in this study had to meet all inclusion criteria: (1) lactating cows; (2) reproductive data were routinely recorded in the HHPS database; and (3) cows were treated with hormone treatment.

The following information was available for each cow on the day of hormone treatment: cow ID, parity, BCS based on the scoring system of Ferguson et al. [29] (<3.00 or >3.75 and 3.00–3.75), DIM, AI history (yes and no), history of hormone use (yes and no), season (winter = 1 January to 2 March, summer = 3 March to 25 May, and rainy = 26 May to 31 August), herd ID, herd size (20 lactating cows and >20 lactating cows), housing system (tie-stall and loose housing), clinician experience (<6 years and 6 years), estrus expression after treatment (yes and no), and AI event after treatment (yes and no). Additionally, ovarian structure data at the initial time of hormone treatment were retrieved and categorized as cows with CL present at the beginning of hormone treatment (cyclic cow) and cows with inactive ovaries and without CL present (noncyclic cow). Furthermore, pregnancy success data after treatment (non-pregnant and pregnant) were also available for each cow. Cows were considered as non-pregnant if they met at least one of the following criteria: (1) cows were diagnosed as non-pregnant; (2) cows returned to estrus before pregnancy diagnosis; (3) cows were re-inseminated before pregnancy diagnosis; and (4) there was no AI event after treatment. The records were also excluded if pregnancy data were missing.

We aimed to study the performance and cost-efficiency of single-treatment hormonal therapy in anestrous dairy cattle in Thailand. We did not aim to include an analysis of repeated treatment protocols (that is, repeated treatments with hormonal therapy of the same animal throughout lactation period) in this study. Additionally, to assess common practices, hormone treatment protocols with less than 2% of the total records were excluded. Data on hormone treatment protocols were also collected and explored for validity and reliability; incomplete and incorrect protocols were excluded. Then, the remaining protocols were categorized based on an estrus detection requirement before AI: (1) estrus detection before AI (EAI) and (2) fixed-time AI (TAI) (Figure 1). Only gonadotropin-releasing hormone (GnRH; buserelin) and prostaglandin F2α (PGF2α; cloprostenol) were used to treat anestrous dairy cows during the study period.

Regarding cost-efficiency analysis, total treatment cost was calculated as the sum of hormone cost, veterinary service fee, and frozen semen and AI costs. Costs associated with routine daily management were not included in the analysis, including labor and estrus detection, and monthly costs of herd health service. All costs were estimated using the average price in the Thai market quoted here in US dollars. The exchange rate from 8 May 2020 was used (USD 1 = THB 32.28) for the whole analysis [30]. The costs of hormone drugs per single dose used in the dairy population studied were USD 6.20 for buserelin (Receptal^®^, Intervet International GmbH, Unterschleißheim, Germany) and USD 2.48 for cloprostenol (Estrumate^®^, Rahway, NJ, USA). The veterinary service fee was estimated to be USD 3.10 per cow for the entire treatment program. The frozen semen and artificial insemination cost was calculated based on the regular price reported in the study area (USD 3.10 per service).

### 2.3. Statistical Analysis

Data analysis was carried out using the STATA version 16.0 statistical software package (Stata Corp., College Station, TX, USA). Estrous expression, AI event, and pregnancy success after hormone treatment were used to calculate estrus detection risk, insemination risk, pregnancy risk, and pregnancy per AI. Estrus detection risk (ER), the probability of estrus detection after treatment, was calculated by the number of cows detected in estrus after treatment divided by the number of cows treated that were supposed to be detected in estrus after treatment. Insemination risk (IR), the probability of AI after treatment, was calculated by the number of cows inseminated after treatment divided by the number of cows treated that were supposed to be inseminated after treatment. Pregnancy risk (PR), the probability of pregnant cows after treatment, was calculated as the number of cows diagnosed as pregnant with AI after treatment divided by the number of cows treated that were supposed to be detected in estrus and inseminated after treatment. Pregnancy per AI was calculated as the number of cows diagnosed as pregnant with AI after treatment divided by the number of inseminated cows after treatment. The chi-square test (*p* < 0.05) was used to compare the ER, IR, and PR of two types of hormone protocols between cyclic and noncyclic dairy cows.

The success of pregnancy was considered the outcome of interest for the multivariate logistic regression analysis. The analysis evaluated the influence of hormone treatment protocols on pregnancy success after treatment. All explanatory variables are shown in the causal diagram (Figure 2). Descriptive statistics (mean, standard deviation, minimum, maximum, frequency, and percentage) were used to assess the characteristics of the outcome variable and all explanatory variables. Explanatory variables with missing values of more than 10% were excluded from the analysis. Continuous variables were assessed for linearity on a logit scale using the Lowess smoothing curve, Box–Cox, and fractional polynomials [31]. In addition, the correlation among the independent variables was assessed using Pearson’s correlation, Kruskal–Wallis test, and the chi-square test. Unconditional association was tested between explanatory variables and outcome, and variables that were unconditionally associated (*p* < 0.20) with outcome were retained for use in the multivariate logistic regression model. A manual stepwise backward approach was applied to estimate the final multivariate model; The approach was initialized using all significant explanatory variables from the univariate analysis. The variables with the highest *p*-values (*p* > 0.05) were removed with the addition of previously removed variables. A variable was considered a confounder and retained in the final model when the removal variables altered the coefficient of the remaining variable by more than 20% and met the following criteria: (1) was associated with the outcome; (2) was not the consequence of the outcome; and (3) was not an intervening variable in the causal diagram. However, nonsignificant variables that were not confounders were permanently removed, and the final multivariate model was decided when all explanatory variables in the model showed a significance level (*p* < 0.05). The biologically plausible interactions between the explanatory variables were tested and kept in the model if they were significant (*p* < 0.05). The fit of the final model was tested using the Hosmer–Lemeshow goodness-of-fit test. In addition, the sensitivity, specificity, and area under the ROC curve (AUC) of the model were calculated to estimate the predictive ability of the model. DeltaX^2^ and Delta-beta were used to identify outliers. The outliers were removed from the dataset and investigated for their effect on our results.

The total cost of treatment per cow and the total cost per pregnancy were calculated for TAI and EAI among cyclic and noncyclic dairy cows. The total treatment cost per cow per protocol was calculated as the total treatment costs for a particular protocol (Figure 1) on all farms, divided by the number of cows treated in that protocol. The total cost per pregnancy was calculated similarly to the total cost of treatment per protocol divided by the number of pregnancies achieved in that protocol. A two-sample independent *t*-test (*p* < 0.05) was used to compare the total treatment cost, hormone cost, veterinary service fees, and frozen semen and AI costs of two types of hormone protocols between cyclic and noncyclic dairy cows.

## 3. Results

### 3.1. Study Population

A total of 279 records of 279 cows on 64 farms were included in the final dataset for data analyses. Seven treatment protocols (Figure 1) were classified as (1) estrous detection before AI (EAI; Protocols 1–4) and (2) fixed-time AI (TAI; Protocols 5–7). Thirty-eight farms (66% of cows), twenty-two farms (27% of cows), and four farms (7% of cows) were in Ratchaburi, Kanchanaburi, and Nakhon Pathom provinces, respectively. A total of 35 farms (62% of cows) were registered as employing a loose-housing system, and 29 farms (38% of cows) used tie-stalls.

### 3.2. Hormone Treatment and AI Outcomes

The successes of single hormone treatment protocols and AI following observed estrus or timed AI are shown in Table 1. The pregnancy risk for the TAI protocol was higher than that for the EAI protocol, but pregnancy per AI did not differ significantly between the two protocols in cyclic and noncyclic cows.

### 3.3. Factors Associated with Pregnancy Success after Hormonal Treatment

The distribution of the level variables of interest and the univariate analysis can be found in Table 2. According to the diagram in Figure 2, the ovarian structure at the initial time of hormone treatment and season were considered confounding factors for the hormone treatment protocol in the final model. Although the ovarian structure at the initial time of hormone treatment was not related to pregnancy outcome in the univariate analysis, it was forced into the multivariate model to control for the known effect, explained by the previous study [12]. As a result, the coefficient of the hormone treatment protocol was changed by more than 30%. Although the history of AI was related to pregnancy outcome in the univariate analysis, it was highly correlated with the season. Moreover, only the coefficient of the season variable was affected and became nonsignificant when AI history and season remained in the final model. Therefore, the history of AI was considered the intervening variable for the season and was removed from the analysis. There was no interaction between explanatory variables found in this study. Farms and clinicians were not significant when accounting for them as random effects in the multilevel logistic regression model. The final results of the multivariate model for factors associated with pregnancy success are presented in Table 3.

The goodness of fit of the final model was evaluated using the Hosmer–Lemeshow test and revealed very weak evidence of lack of fit (*p* = 0.85). The sensitivity, specificity, and AUC were 0.70, 0.68, and 0.72, respectively. The cut-off point for the sensitivity and specificity measures was 0.18, resulting from a sensitivity–specificity plot. Evaluating outliers and influential observations did not reveal weaknesses in the model.

### 3.4. Cost-Efficiency for Hormonal Treatment

The total cost for the TAI protocol was higher than that for the EAI protocol in cyclic and noncyclic cows; however, the cost per pregnancy for the TAI protocol was lower than that for the EAI protocol in noncyclic cows (Table 4).

## 4. Discussion

### 4.1. Hormone Treatment and AI Outcomes

The PR for the TAI protocol was significantly higher than that for the EAI protocol in noncyclic cows; there was a trend towards significance in cyclic cows. However, when cows were inseminated after treatment, there were no differences in pregnancy according to AI between the EAI and TAI protocols for both noncyclic and cyclic cows, which was in accordance with previous studies [32,33]. The low PR for the EAI protocol in both cyclic and noncyclic cows can be explained by the fact that the IR for the EAI protocol was significantly lower than that for the TAI protocol because the IR for the EAI protocol relies on ER that requires estrus detection by the farmer. In contrast, all cows that received the TAI protocol were inseminated after hormone treatment whether they showed estrous behavior or not. The results showed that the ER for the EAI protocol was low, especially in noncyclic cows, which can be explained by limited knowledge of estrus detection among Thai dairy farmers [34]. Therefore, the majority of cows that received the EAI protocol lost the opportunity to be inseminated after hormone treatment. Moreover, GnRH analogs have a limited half-life which does not exceed 5–6 h, which represents the main side effect of the GnRH analog [21]. Therefore, its use in the EAI protocol for noncyclic cows may not be efficient for estrous behavior because the estrus induction efficiency varies and depends on the size and stage of ovarian follicles at the time of treatment [21]. However, Amin and Said [21] reported that adding chitosan to GnRH increases its efficacy, accelerates the growth of ovarian follicles, and improves hormonal insufficiency, resulting in an increased estrous induction rate and a conception rate. Therefore, GnRH-conjugated chitosan might be an appropriate choice for application in the EAI protocol for noncyclic cows. Additionally, the administration of GnRH at the time of AI may increase ovulation and pregnancy risk in both noncyclic and cyclic cows [35].

ER was higher in cyclic cows than noncyclic cows following the EAI protocol due to the severity of anestrous. Previous research showed that cyclic cows display obvious expression of estrous behavior and a longer estrus duration than noncyclic cows [33]. This phenomenon can be explained by the concept of the depth of anestrous; cows in deep anestrous conditions are more affected by many factors that have a deleterious effect on ovarian activity and are more difficult to treat [36]. Noncyclic cows suffering from true anestrous or inactive ovaries had deeper anestrous than cyclic cows and rarely showed estrous behavior after treatment. It was surprising that the ER for the TAI protocol was limited to about 0.3 in both cyclic and noncyclic cows, which is noticeably low. This might be because farmers may ignore estrus detection when TAI protocols are applied while paying more attention to cows treated with EAI protocols. Another reason could be the consequence of the last injection of GnRH before AI in the TAI protocol [33] (Figure 1). GnRH injection suppresses estrous behavior because GnRH directly induces the release of LH and ovulation before the peak of estrogen. Therefore, estrus behavior would be less noticeable or may not occur before AI when the TAI protocol was applied [37,38,39]. However, in both noncyclic and cyclic cows regardless of the hormone treatment protocols, some animals might have silent ovulation, where ovulation occurs without showing any estrus signs and leads to low ER [40,41]

### 4.2. Factors Associated with Pregnancy Success after Hormonal Treatment

When multivariate logistic regression analysis was performed to evaluate factors associated with pregnancy success, cows that received the TAI protocol were about four times more likely to become pregnant than cows that received the EAI protocol. This is consistent with our study results mentioned above: The TAI protocol showed a higher proportion of pregnant cows than the EAI protocol in both noncyclic and cyclic cows.

Cows with 3.00 ≤ BCS ≤ 3.75 were more likely to become pregnant than cows with BCS < 3.00 or BCS > 3.75. This result was consistent with previous studies reporting that cows with a BCS ≥ 3.00 are more likely to conceive and remain pregnant than cows with a BCS < 3.00 following hormone treatment [42,43]. Our study proposes that this could be the long-term adverse effect of NEB from the transition period [44], especially in Thailand, where dairy farmers are not well trained in nutritional management [45]. Unfavorable metabolic changes during NEB have many detrimental effects on the quality of oocytes and embryos, resulting in unsuccessful fertilization and pregnancy after insemination [46]. Low BCS has an adverse impact on follicle development and reduces ovulation rate after hormone treatment [47]. Furthermore, Madureira et al. [48] reported that estrous expression was affected by BCS, since cows with BCS ≤ 2.50 expressed less intense estrous activity. This disturbed estrus detection efficacy when the EAI protocol was applied, resulting in estrus detection failure and loss of AI opportunities. On the other hand, the lower pregnancy success among cows with BCS > 3.75 compared to cows with 3.00 ≤ BCS ≤ 3.75 can be explained by the fact that cows with BCS > 3.75 can present insulin-resistant complications; there is evidence that this effect results in reduced oocyte quality and impaired embryo development [49].

The loose-housed cows had better pregnancy success than the tie-stall cows after hormone treatment. This finding is consistent with previous research indicating that loose-housed cows showed better fertility than tethered cows; loose-housed cows have a better pregnancy rate after first AI, number of services per conception, calving to conception interval, and calving interval [50,51]. Additionally, loose-housed cows can freely show estrous behavior and provide greater opportunities to realize estrous behavior, while tie-stall housing systems greatly impair estrous expression because cows are restricted to individual stalls [52]. Movement restriction in the tie-stall system might have negative effects on reproductive function due to physiological stress; tie-stall cows had a higher concentration of hair cortisol compared to loose-housed or free-stall cows [53,54].

The season negatively impacted pregnancy success after hormone treatment. Pregnancy probabilities increased from the rainy season to the summer season and reached the highest point in the winter. Our findings agreed with the most recent Thai report that the probability of pregnancy in the winter season was higher than in the rainy season and the summer season after hormone implementation [13]. A low proportion of pregnancies has also been reported in bovines after receiving hormone treatments during the rainy season in Thailand [42]. Biological reasons for this can be explained by the effect of heat stress; cows usually face mild stress conditions in the winter season (THI value of around 77), after which they experience moderate heat stress during the summer and rainy seasons (THI values of around 82 and 81, respectively) [13]. Heat stress adversely affects fertility by altering reproductive physiology through hyperthermia, oxidative stress, hormone imbalance, and negative energy balance [19,55,56].

The effects of heat stress result in reduced oocyte quality, decreased semen quality, and impaired embryo development, which contribute to poor estrous expression and infertility, especially in Thailand [57]. Furthermore, our study result showed that pregnancy success reached the lowest point in the rainy season. Schüller et al. [58] reported that the likelihood of dairy cows becoming pregnant is reduced by both short- and long-term heat stress. Long-term heat stress before the day of breeding leads to a reduced number of follicles, a reduced estradiol concentration in the follicles, and an earlier emergence of the dominant follicle [59,60]. In our study, we proposed that anestrous cows received hormone treatment in the rainy season; these cows experienced heat stress during the summer season, resulting in low pregnancy success after treatment. Additionally, other environmental factors, such as rainfall, also affected pregnancy success. Heavy rain, strong wind, or high humidity indirectly affect pregnancy success due to suppressing estrous behaviors [61,62], which will affect estrus detection efficiency and AI timing, especially in cows receiving the EAI protocol.

The season and ovarian structure at the initial time of hormone treatment were also identified as confounding variables for the hormone treatment protocol. The reason for this is the decision of clinicians, who were more likely to treat anestrous cows with the EAI protocol rather than the TAI protocol in winter, contrasting with the summer and rainy seasons due to the adverse effect of heat stress and economic concerns. Similarly, clinicians might also be aware of the low estrous expression ability in noncyclic cows [32]. Therefore, far more cyclic cows were treated by the EAI protocol than noncyclic cows, and a majority of noncyclic cows were treated by the TAI protocol (Table 1).

The findings of our study did have some limitations. First, enrollment in the study was selected from the hospital database using the anestrous definition and selection criteria. Therefore, there was the potential for selection bias to exist because the study population was not randomly selected from the target population. Second, misclassification of the ovarian structure at the initial time of treatment may have occurred due to the misidentification between CL and follicles by clinicians and due to the recall bias by farmers when reporting estrous expression. All anestrous cows were diagnosed and treated by six different clinicians in this study, and the cows came from different farms. However, due to the limited small sample in each category, the mixed model cannot account for farm and clinicians as a random effect. This study assumed that the sources of errors were equally distributed. Finally, the AUC value revealed that the overall diagnostic precision of the model was acceptable [30]. The probability that the model correctly classified pregnancy success after hormone treatment was 72%.

### 4.3. Cost-Efficiency for Hormonal Treatment

The total average treatment cost per cow indicated that the TAI protocol was more expensive than the EAI protocol due to the higher cost of hormones and frozen semen and the AI service. The cost of hormones represented the majority of the treatment costs, the most expensive of which was GnRH. Ricci et al. [24] reported that the GnRH expense had a greater impact on the net profit gain than prostaglandins. The average cost of frozen semen and AI service was higher than that of the EAI protocol. These findings also support the observation we initially reported that the cost of hormone therapy is a substantial component of total costs, probably contributing to the erroneous belief of non-adopter Thai dairy farmers that it may not be cost-effective.

If only the total average treatment cost per cow is taken into account, fixed-time AI hormonal treatment appears more expensive than estrus detection alone. However, one must also take into account the resulting pregnancies that arise due to treatment. The cost per pregnancy for noncyclic cows receiving the EAI protocol was about twice as much compared to the cost for the TAI protocol. The PR for the EAI protocol was lower than that for the TAI protocol. However, the cost per pregnancy for cyclic cows receiving the EAI protocol was lower than that for the TAI protocol; the EAI protocol tended to have a lower PR than the TAI protocol. We propose that the TAI protocol is the most cost-efficient single-treatment hormonal therapy option for noncyclic cows, whereas the EAI protocol is the most cost-efficient option for cyclic cows.

We note that we were unable to collect a set of farm economic data as we would ideally have liked. Under field conditions, farmers rarely recorded milk production costs and income. Our calculations reflect hormone treatment costs without taking into wider consideration other production costs including labor costs, maintenance costs for non-pregnant cows (e.g., feed and feeding cost, and re-insemination cost), culling costs, and the cost of lost milk revenue due to extended periods of nonpregnancy.

## 5. Conclusions

Anestrous is a major reproduction problem in the central-western region of Thailand. This is the first study to integrate the evaluation of hormone treatment outcomes, factors associated with pregnancy success after hormone treatment, and cost-efficiency analysis. These would help farmers and clinicians make the most effective decision to deal with the anestrous problem in dairy cows under tropical conditions, such as in Thailand. Pregnancy success was influenced by the hormone treatment protocol, BCS at the initial time of hormone treatment, housing conditions, and season. The TAI protocol seems to be a very useful method for solving anestrous problems in dairy cows due to high PR. But cost-efficiency analysis showed that the TAI protocol suits noncyclic cows, whereas the EAI protocol suits cyclic cows due to the lower cost per pregnancy. Therefore, the hormone treatment decision to solve the anestrous problem should be based on the health of the animal, environmental conditions, farm and management conditions, and the cost-effectiveness of treatment protocols.

## Figures and Tables

**Figure 1 animals-14-01564-f001:**
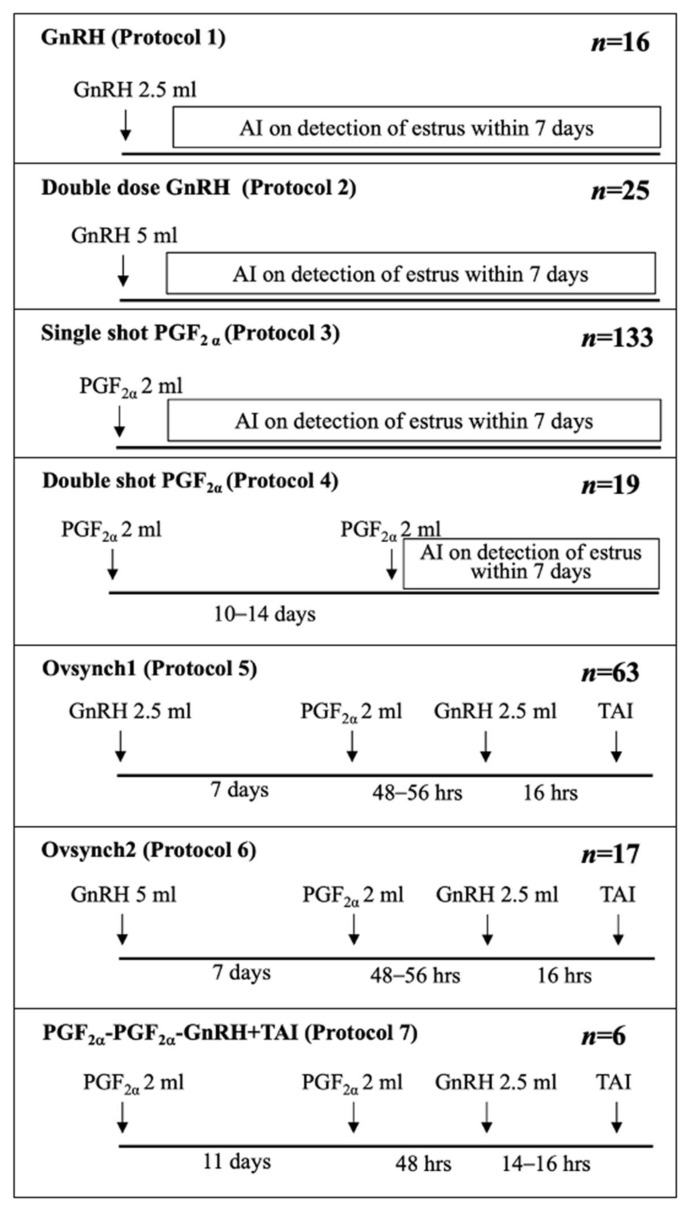
Schematic diagram showing hormone protocols that were applied among 279 anestrous dairy cows from 64 herds under the Herd Health and Production Service Program, the Large Animals Unit, Veterinary Teaching Hospital, Nong Rho, Kasetsart University, in 3 provinces of the central-western region of Thailand from January to August 2017; Protocols 1 to 4 were recognized as hormone treatment with estrus detection before AI (EAI), and Protocols 5 to 7 were recognized as hormone treatment with fixed-time AI (TAI). TAI = fixed-time artificial insemination, AI = artificial insemination, GnRH = gonadotropin-releasing hormone, PGF_2α_ = prostaglandin F2alpha.

**Figure 2 animals-14-01564-f002:**
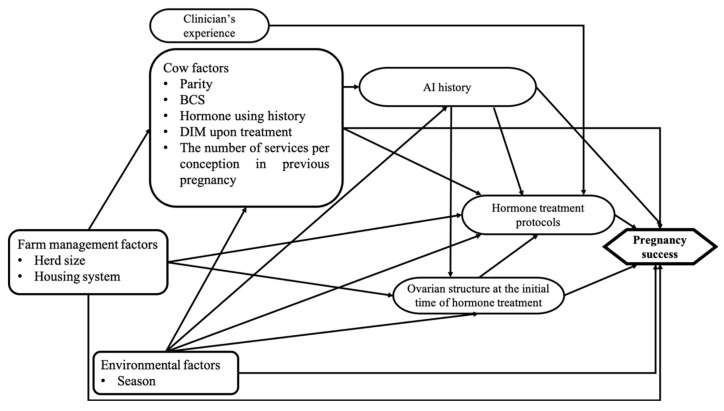
Causal diagram illustrating postulated causal paths linking all explanatory variables related to pregnancy success in anestrous dairy cows that were treated by hormone treatment protocols and raised under tropical conditions.

**Table 1 animals-14-01564-t001:** Hormone treatment and AI successes for anestrous dairy cows based on ovarian structure at the initial time of hormone treatment and treatment protocols among 64 herds in the central-western region of Thailand in 2017.

Treatment Response	Noncyclic		*p*-Value	Cyclic		*p*-Value
	EAI (*n* = 41)	TAI (*n* = 64)		EAI (*n* = 152)	TAI (*n* = 22)	
Estrus detection risk	0.24	0.30 ^1^	0.690	0.74	0.38 ^2^	0.001
Insemination risk	0.24	1.00	<0.000	0.66	1.00	0.003
Pregnancy risk	0.07	0.22	0.048	0.17	0.32	0.099
Pregnancy per AI ^3^	0.29	0.21	0.570	0.26	0.31	0.578

^1^ Number superscripts indicate there are 5 missing data. ^2^ Number superscripts indicate there are 1 missing data. ^3^ Pregnancy per AI was defined as the rate of cows that became pregnant after AI following hormone treatment. Noncyclic = cows with the absence of visible estrus detected by farmers for ≥30 days before the day of hormone treatment or diagnosed by clinicians, and with inactive ovaries and no CL present; Cyclic = cows with the absence of visible estrus detected by farmers for ≥30 days before the day of hormone treatment or diagnosed by clinicians, and with CL present; EAI = hormone treatment with estrus detection before AI; TAI = hormone treatment with fixed-time AI.

**Table 2 animals-14-01564-t002:** Descriptive statistics for hormone treatment protocols and other exploratory variables that measured the unconditional association with pregnancy success at 60–90 days following AI after treatment (*p* ≤ 0.2) among 279 anestrous dairy cows from 64 herds in the central-western region of Thailand in 2017.

Variable	Category	N	% or Mean ± S.D.		*p*-Value
			Pregnant	Non-Pregnant	
Hormone treatment protocols	EAI	193	24.42	84.97	0.064
	TAI	86	15.03	71.62	
Ovarian structure ^1^	Noncyclic	105	16.19	83.81	0.556
	Cyclic	174	18.97	81.03	
AI history	Yes	145	23.45	76.55	0.013
	No	134	11.94	88.06	
Hormone-use history	Yes	28	14.29	85.71	0.598
	No	251	18.33	81.67	
Housing system	Tie stall	105	11.43	88.57	0.024
	Loose housing	174	21.84	78.16	
Herd size	≤20	111	12.61	87.39	0.061
	>20	168	21.43	78.57	
Season ^1^	Winter	83	24.10	75.90	0.148
	Summer	95	17.89	82.11	
	Rainy	101	12.87	87.13	
BCS ^1,2^	<3.00 or >3.75	237	16.03	83.97	0.152
	3.00–3.75	42	28.57	71.43	
Day in milk ^1^	Continuous	279	220.00 ± 115.00	200.00 ± 134.00	0.073
Parity	Continuous	279	3.00 ± 2.00	3.00 ± 2.00	0.699

Noncyclic = cows with the absence of visible estrus detected by farmers for ≥30 days before the day of hormone treatment or diagnosed by clinicians, and with inactive ovaries and no CL present; Cyclic = cows with the absence of visible estrus detected by farmers for ≥30 days before the day of hormone treatment or diagnosed by clinicians, and with CL present; EAI = hormone treatment with estrus detection before AI; TAI = hormone treatment with fixed-time AI; Coeff. = coefficient. ^1^ Recorded at the initial time of hormone treatment. ^2^ Body condition score on a 1–5 scale [29].

**Table 3 animals-14-01564-t003:** Results from the final multivariable logistic regression used to investigate the relationship between exploratory variables and pregnancy success at 60–90 days following hormone treatment and AI among 279 anestrous dairy cows from 64 herds in the central-western region of Thailand in 2017.

Variable	Category	Coeff.	SE	Odds Ratio	95% CI	*p*-Value
Hormone treatment protocols	EAI	Ref.				0.002
	TAI	1.30	0.42	3.68	1.60–8.50	
Ovarian structure ^1^	Noncyclic	Ref.				0.182
	Cyclic	0.56	0.42	1.76	0.76–4.00	
BCS ^1^	<3.00 or >3.75	Ref.				0.048
	3.00–3.75	0.82	0.41	2.28	1.00–5.15	
Housing system	Tie stall	Ref.				0.010
	Loose housing	0.99	0.38	2.70	1.27–5.76	
Season ^1^	Winter	Ref.				
	Summer	−0.88	0.42	0.41	0.18–0.93	0.035
	Rainy	−1.27	0.44	0.27	0.11–0.66	0.004

Noncyclic = cows with the absence of visible estrus detected by farmers for ≥30 days before the day of hormone treatment or diagnosed by clinicians, and with inactive ovaries and no CL present; Cyclic = cows with the absence of visible estrus detected by farmers for ≥30 days before the day of hormone treatment or diagnosed by clinicians, and with CL present; EAI = hormone treatment with estrus detection before AI; TAI = hormone treatment with fixed-time AI; Coeff. = coefficient. ^1^ Recorded at the initial time of hormone treatment.

**Table 4 animals-14-01564-t004:** Hormone treatment costs for anestrous dairy cows based on ovarian structure at the initial time of hormone treatment and treatment protocols among 279 anestrous dairy cows from 64 herds in the central-western region of Thailand in 2017.

Parameter	Noncyclic		*p*-Value	Cyclic		*p*-Value
	EAI	TAI		EAI	TAI	
Hormone cost per cow (USD)	9.98 ± 0.50	16.33 ± 0.33	<0.01	2.79 ± 0.07	14.43 ± 0.58	<0.01
Frozen semen and AI service fees per cow (USD)	0.53 ± 0.18	2.42 ± 0.16	<0.01	1.51 ± 0.13	2.11 ± 0.32	0.09
Veterinary service fees per cow (USD)	3.10	3.10	>0.05	3.10	3.10	>0.05
Total treatment cost per cow (USD)	13.60 ± 0.5	21.86 ± 0.36	<0.01	7.40 ± 0.15	19.64 ± 0.60	<0.01
Cost per pregnancy (USD)	186.44 ± 6.94	99.79 ± 1.64	-	43.27 ± 0.88	61.75 ± 1.89	-

Noncyclic = cows with the absence of visible estrus detected by farmers for ≥30 days before the day of hormone treatment or diagnosed by clinicians, and with inactive ovaries and no CL present; Cyclic = cows with the absence of visible estrus detected by farmers for ≥30 days before the day of hormone treatment or diagnosed by clinicians, and with CL present; EAI = hormone treatment with estrus detection before AI; TAI = hormone treatment with fixed-time AI.

## Data Availability

The data that support the findings of this study are available from the authors, but restrictions apply to the availability of these data, which were used under permission from the Faculty of Veterinary Medicine, Kasetsart University. Data are, however, available from the authors upon reasonable request.

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
