# Peer review of "Performance and Cost-Efficiency of Single Hormonal Treatment Protocols in Tropical Anestrous Dairy Cows"

_animals, 2024, doi:10.3390/ani14111564_

Round 1

Reviewer 1 Report

Comments and Suggestions for Authors

The manuscript aimed to evaluate the performance of the hormone treatment  protocols, determine the factors associated with pregnancy success after hormone treatment, and  compare the cost-efficiencies of two types of hormone treatment among cyclic and noncyclic anestrous dairy cows. The cost-efficiency analysis showed that the TAI protocol was the most cost-efficient option for noncyclic cows, whereas the EAI protocol was the most cost-efficient option for cyclic cows.

Strong points:- the study was carried out on a large number of cows;

                        -the study presents the statistical analysis of the data;

                        -the study presents the cost per pregnancy ($) for the EAI(hormone treatment with estrus detection before AI) and TAI(hormone treatment with fixed-time AI) protocols for both cyclic and non-cyclic cows.

                        - the conclusions are clear and in accordance with the presented results.

Weak points: -The selection of cows for both protocols was based on the diagnosis established by veterinarians who did not directly participate in this study

                        - The methods by which the diagnosis of anestrus was established and the ovarian diseases of the cows taken in the study are not presented

                        - The animals taken in the study were diagnosed and treated by six different clinicians

                        - Within the EAI protocol, estrus detection was performed subjectively by the owners

Although some of the references on which the manuscript is based include journals that are not indexed in SCImago or Science Report, the subject of the manuscript is still of great relevance from a clinical point of view and can be published.

Reviewer 2 Report

Comments and Suggestions for Authors

This article is well written and I think there is no major problem. Consider to correct several minor points noted below.

The schema is described in Figure 2, but it is somewhat confusing. Based on the results of the experiment, it should be revised to emphasize which factors affect Pregnancy success, and it should be at the end of the paper.

It should be discussed about the reason why Estrus detection risk of Non-cyclic cows is very low in Table2.

There are lot of protocols in Figure7. Is there difference in ER, IR and PR during each protocol?

(Line 367) [33 should be [33].

Reviewer 3 Report

Comments and Suggestions for Authors

Dear authors,

Anestrus syndrome represents a commonly encountered issue in bovine pathology, and various hormonal protocols are always welcome to be studied so that the veterinarian can offer the most practical solutions to farmers in order to streamline zootechnical activities efficiently. Providing real data about the costs of different protocols represents a major benefit useful for both farmers and veterinarians. Single hormonal treatments alone seem to be the solution most of the time in treating anestrus syndrome, which is why the need to conduct such a study is imperative.

Given the topic's high relevance in the day-to-day practice of veterinary doctors working in the field of reproduction, the article should be easily understandable and provide real, easily comprehensible data. Therefore, suggestions will be made for each subchapter with the aim of achieving these goals.

The introduction is well-written, encompassing data that gives the reader an idea about the general thesis. Perhaps a paragraph about the economic impact of administering different hormones would be welcomed.

Materials and Methods: This section is not easily understandable and requires increased attention from the reader, thus it needs to be rewritten. Considering the low incidence, protocols 8, 9, and 10 are no longer relevant and should be excluded from the study. Protocols need to be briefly described, and the economic efficiency aspect should be practically analyzed for each protocol, as they essentially make the economic difference.

Results. The results should be strictly based on the animals included in the study; those excluded for various reasons should no longer be mentioned. Additionally, the results need to be rewritten to provide real data about each protocol used and to highlight the economic benefits and limitations of each one. At the end of each protocol, the reader should learn the implementation cost and the success rate.

The discussion and conclusions section also needs to be rewritten, including a comparison between protocols regarding both the gynecological results obtained and the financial factor.

      Comments on the Quality of English Language

The article is writed in a coherent manner, exhibiting linguistic clarity that renders it readily comprehensible.

Round 2

Reviewer 3 Report

Comments and Suggestions for Authors

Dear authors,

I believe that the article has undergone considerable improvement following the latest revisions and is now suitable for publication in its current state. However, I'm leaving here a study published in Frontiers in 2023, "Economical implications and the impact of gonadotropin-releasing hormone administration at the time of artificial insemination in cows raised in the extensive system in North Romania," which you may consider for the discussion section.

Best regards!

Comments on the Quality of English Language

Minor editing of English language required
